# Ultrasonic Regeneration Studies on Activated Carbon Loaded with Isopropyl Alcohol

**Hsuan-Yi Hong, Niels Michiel Moed \*, Young Ku and Hao-Yeh Lee**

Department of Chemical Engineering, National Taiwan University of Science and Technology,
Taipei City 10607, Taiwan; elfsj198@gmail.com (H.-Y.H.); ku508@mail.ntust.edu.tw (Y.K.);
haoyehlee@mail.ntust.edu.tw (H.-Y.L.)
\* Correspondence: nikolai.moed@gmail.com; Tel.: +886-(0)-970137181



**Featured Application: Ultrasonic regeneration was applied to regeneration of activated carbon loaded with isopropyl alcohol**

**Abstract:** Ultrasonic regeneration of activated carbon loaded with isopropyl alcohol (IPA) was studied. IPA adsorption was performed batchwise at varying solution pH. Adsorption was optimal at solution pH 7, which was closest to the point of zero charge of the activated carbon (6.7). Ultrasonic regeneration was performed on IPA-loaded activated carbon with three factors being tested: ultrasonic intensity, solution temperature and ethanol addition. Regeneration efficiency increased with ultrasonic intensity up to 32.4 W/cm$^2$. A higher intensity led to a higher desorption but damaged the activated carbon, shown by a decrease in the particle size of activated carbon. The regeneration efficiency increased with solution temperature primarily because desorption is endothermic and because the surface tension and viscosity of a solution are reduced with increasing temperature, promoting cavitation bubble production. Ethanol addition increased regeneration efficiency up to 10%, as ethanol reduces tensile stress, facilitating cavitation bubble generation. At 15% and above, regeneration decreases, possibly due to coalescence of bubbles into larger, more stable bubbles. Under optimal parameters, the regeneration efficiency was 83%, which dropped to 64% after four regeneration cycles.

**Keywords:** activated carbon; ultrasonic; regeneration; IPA; isopropyl alcohol; adsorption

## 1. Introduction

The electronics industry is an important industrial sector which generates wastewater containing various organic solvents. This includes isopropyl alcohol (IPA), which is toxic by inhalation, ingestion or absorption and should be removed or reclaimed before discharge into the environment [1,2]. Activated carbon, due to its microporous character, high surface area and the chemical nature of its surface, is commonly used as an adsorbent for organic pollutants, such as IPA, in low concentrations [3]. Different factors influence the adsorption capacity and adsorption equilibrium, such as the nature of adsorbent, the nature of the adsorbate and the solution composition. Solution pH influences the adsorption to a large extent, as it affects the properties of both adsorbent and adsorbate. The initial concentration of adsorbate plays a significant role in the adsorption, providing the driving force to overcome mass transfer resistance. As the activated carbon adsorbs more and more adsorbate, it reaches a point where it is saturated. The saturated activated carbon is then frequently buried in a landfill, incinerated, or regenerated. Strict regulations for emission control drive a rising demand for activated carbon in environmental applications. The global market size of activated carbon was 2.7 million tons with a value of 4.7 billion USD in 2015 and is projected to reach 5.4 million tons with

a value of 8.1 billion USD by 2021 [4]. To produce activated carbon, a large amount of biomass is needed, with 6.7 tons of coconut shell required to produce 1 ton of activated carbon [5].

Considering this high biomass requirement and rising market demand, regeneration is economically and environmentally beneficial as it can greatly reduce the need for the manufacturing of new activated carbon. This regeneration is achieved through a variety of methods which can be categorized as desorption and decomposition. Desorption of activated carbon can be thermal (using steam, hot water, microwave, etc.) or non-thermal (using solvents, supercritical fluids, etc.) and works through principles that break the bonds between adsorbent and adsorbate. Decomposition of activated carbon works through the breaking down of adsorbate on the surface of adsorbent and is classified into electrochemical, microbiological, chemical and ultrasound (which is also a desorption method) [6]. Ease and practicality differ among techniques but high regeneration efficiencies can be accomplished for practically all methods. Taking phenol as an example, regeneration efficiencies of 90% and above were reached using pyrolysis [7] and bio-regeneration [8] and between 70–80% using electrochemistry [9,10], photocatalysis [11] and ultrasound [12].

Ultrasonic regeneration is a promising technique that offers advantages, such as being safe, clean and energy-conserving [13,14] and being able to desorb a wide range of chemicals. In one study, 64% of adsorbed trichloroethylene was desorbed from granular activated carbon after 1 h of ultrasonic at 20 kHz. [15]. Another study showed the potential in desorption of heavy metals, where 120 min of ultrasonic treatment led to a $Cr^{6+}$ desorption of 45.9%, as opposed to 10.7% desorption in distilled water [16]. One study showed its effectivity for organic solvents, where 70% of phenol was desorbed but BET surface area and total pore volume of the activated carbon was found to decrease by 25% [12]. Ultrasonic regeneration acts through cavitation, which is the formation, growth and implosive collapse of microbubbles. These collapses, which occur in the span of milliseconds, create extreme conditions, such as high temperatures and pressure, which are capable of breaking chemical bonds of water, generating highly active radicals, such as $^\bullet O$, $^\bullet OH$, $^\bullet H$ and $^\bullet HO_2$ [17,18]. This makes ultrasonic regeneration a complicated process, as the effects of ultrasound are both thermal, mechanical and chemical [6]. Many factors influence the ultrasonic regeneration efficiency, such as ultrasonic intensity, solution temperature and addition of additives, such as ethanol. Ultrasonic intensity affects collapse times of microbubbles and the temperature and pressure upon collapse. This in turn, leads to a greater desorption of contaminants [19–22]. There is, however, a potential for damaging the adsorbent by ultrasound, which greatly depends on the ultrasound frequency and intensity. To get an optimum result for regenerating exhausted adsorbent, it is vital to identify the optimal ultrasound intensity [12,18]. In addition to the desorption process being endothermic in nature, higher solution temperatures cause a decrease in a liquid's tensile stress and viscosity, leading to cavitation bubbles being more easily generated [22,23]. The addition of ethanol can reduce the surface tension of aqueous solution, thus reducing the cavitation threshold and facilitating the generation of bubbles. The generation of more transient cavitation bubbles increases the amounts of high-speed microjets and high-pressure shock waves as they collapse. Ultrasound and ethanol even seem to produce a synergistic effect where the overall desorption was found to be greater than the sum of the two separate processes [23]. However, adding excess amounts of ethanol may reduce regeneration efficiency, possibly because of the coalescence of numerous cavitation bubbles, forming larger and more stable bubbles. These stable bubbles dampen the passage of sound energy through the liquid and eliminate many of the smaller bubbles that would have collapsed [20].

The aim of this study was to test the feasibility and effectiveness of ultrasonic regeneration of IPA loaded activated carbon. To properly test desorption, adsorption tests are first performed to identify optimum pH and create a reference point. Optimization for IPA desorption is performed for ultrasonic intensity, temperature and ethanol addition. Finally, a 4-cycle adsorption-desorption test takes place to test the regeneration efficiency over multiple cycles.

## 2. Materials and Methods

### 2.1. Pretreatment and Analysis of Activated Carbon

The activated carbon used in this study was made from coconut shells in Vietnam and supplied by Mega Union Technology, Inc. (Taoyuan, Taiwan). The activated carbon was subjected to acid wash treatment to clear impurities and surface contamination. The activated carbon was immersed in 0.3% HCl solution for 2 h, filtered, washed several times using de-ionized water, dried at 110 °C in an oven until no further weight loss was observed and finally cooled down to room temperature. The physical and chemical characteristics of the activated carbon were determined, with the elemental analysis (EA) and surface elements (EDS) analyzed using a FLASH 2000 elemental analyzer by Thermo Scientific (MA, USA). Surface morphology was examined, using a field-emission scanning electron microscope model JSM-6500F, by JEOL and surface area and porosity (BET) was measured using the Autosorb-1 by Quantachrome (FL, USA). Particle size was measured by sieving the particles and weighing to quantify the fractions. Surface functional groups were tested using Boehm titration and the point of zero charge was determined using the pH drift procedure [24,25].

### 2.2. Adsorption of IPA by Activated Carbon

Adsorption of IPA in aqueous solution was performed in a 500 mL glass beaker equipped with a double cylindrical jacket, an isothermal controller and a thermal couple to keep the solution temperature at 25 ± 1 °C. The reactor was placed on a magnetic stirring plate to provide mixing at 300 rpm. The solution pH was adjusted using HCl and NaOH solutions to a final pH of 3.0, 5.0, 7.0 and 9.0. Other parameters were an initial IPA concentration of 1000 mg/L, 500 mL of solution, 5 g of activated carbon and an operation time of 180 min. The concentration of IPA in solution for all experiments was monitored by taking 1.0 mL aliquots at determined time intervals, which were subsequently analyzed using a 7890A GC-FID with an AB-InoWax capillary column, both by Agilent Technologies (CA, USA).

### 2.3. Ultrasonic Regeneration of IPA-Saturated Activated Carbon

Ultrasonic regeneration of activated carbon saturated with IPA was performed in aqueous solution with a batch system as shown in Figure 1. The reactor itself was similar to the one used in the adsorption experiments. Ultrasonic waves were generated using an ultrasonic processor and probe model JY92-IIN by Scientz. The ultrasonic probe was submerged into the reactor holding 500 mL of distilled water and 10.0 g of IPA-saturated activated carbon. IPA-saturated activated carbon was prepared by mixing 10.0 g of activated carbon in 500 mL containing 1000 mg/L of IPA solution at 25 °C in a sealed conical flask for 24 h. While this does not lead to a concentrating of IPA, the goal of this study was to test the feasibility and the reaction rate of ultrasonic regeneration. During ultrasonic regeneration of the IPA-saturated activated carbon, 1.0 mL samples were periodically taken for IPA concentration analysis using a GC-FID. Upon completion of the ultrasonic regeneration experiment, the activated carbon was separated from the solution by filtration and dried under air for further application. The effects of ultrasonic intensity, water temperature, ethanol addition and regeneration cycles on ultrasonic regeneration were investigated. Particle size of the activated carbon was analyzed to measure the fracture of activated carbon during ultrasonic regeneration. Multiple cycles of adsorption and regeneration were conducted to examine the regeneration efficiency of activated carbon. During optimization the regeneration was quantified by the fraction of IPA desorbed, as measured by GC. In the multiple cycle tests, the amount of IPA adsorbed in the consecutive adsorption is used in comparison to the IPA adsorbed in the initial run.

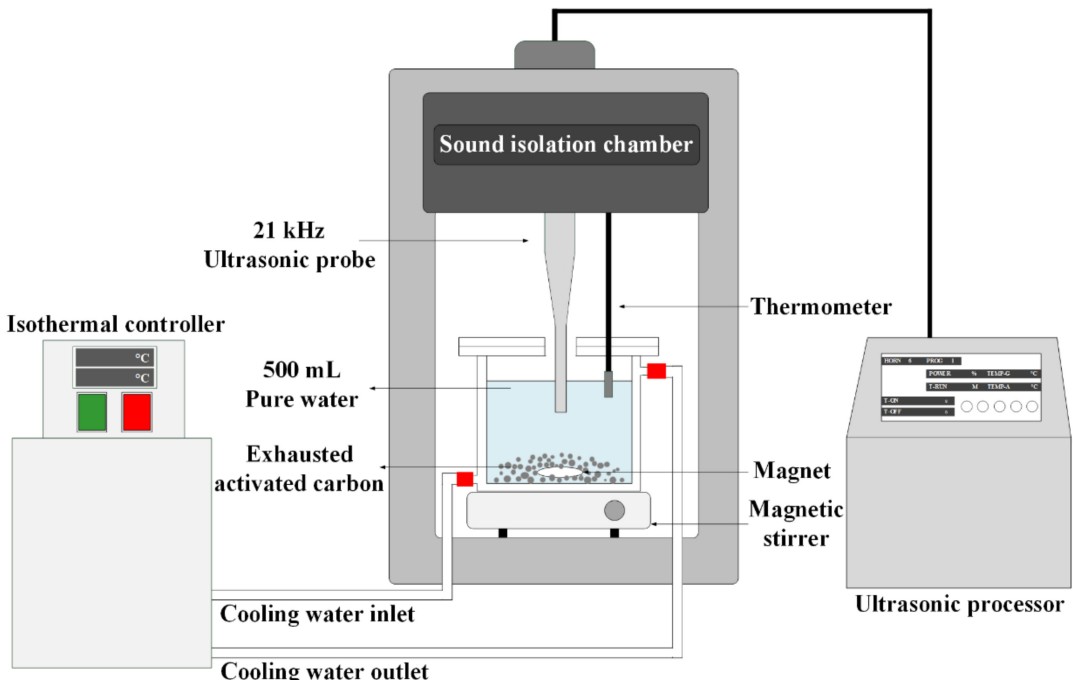

**Figure 1.** Experimental setup for ultrasonic regeneration of exhausted activated carbon.

## 3. Results

### 3.1. Properties of the Activated Carbon

Analysis of surface elements was performed on original (before the acid ash) and fresh (after the acid ash) activated carbon, with the results shown in Table 1. The contaminants of activated carbon were significantly removed by acid washing, as shown by the increased carbon content. As revealed in Figure 2, SEM imaging also revealed that acid washing removed impurities, giving rise to a pore structure with a seemingly smoother surface. The pore structure characteristics of the original activated carbon and fresh activated carbon are presented in Table 2. Fresh activated carbon exhibited a slightly larger specific surface area and total pore volume compared to original activated carbon; this may be an effect of other chemicals, such as oxygen, being removed during the acid washing process. However, the acid washing seems to have little effect on the pore size of activated carbon. The mean pore diameters of the original and fresh activated carbon indicate that a vast majority of the pores fall into the range of micropores. The point of zero charge, which is the pH at which the net surface charge is zero, of fresh activated carbon was found to be about 6.7, as revealed in Figure 3. The surface charge of activated carbon depends on the solution pH and its point of zero charge, the activated carbon surface is positively-charged at pH lower than the point of zero charge and vice versa [26]. Surface functional groups on activated carbon affect the surface charges and adsorption properties of activated carbon. Boehm titration of fresh activated carbon revealed that most of the acidic functional groups are lactone and phenolic groups, followed by the carboxylic group. The total number of the basic functional groups was calculated to be 0.207 mmol/g, which is approximately equal to the total number of acidic functional groups, confirming the previously observed point of zero charge of 6.7.

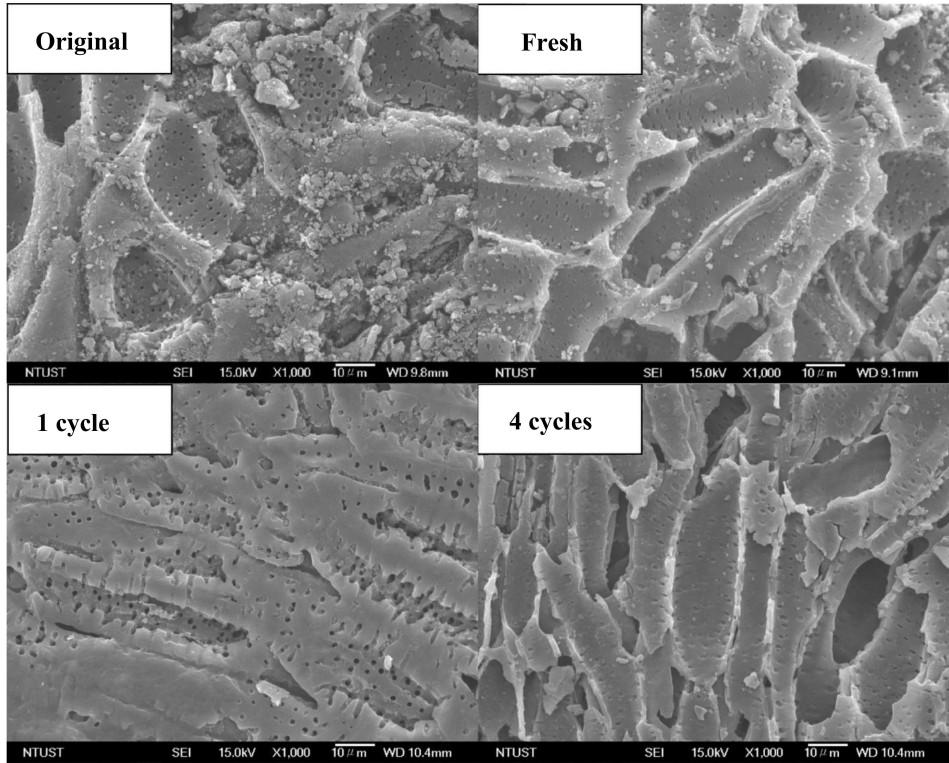

**Figure 2.** SEM images of original, fresh and regenerated activated carbon at 1000× magnification.

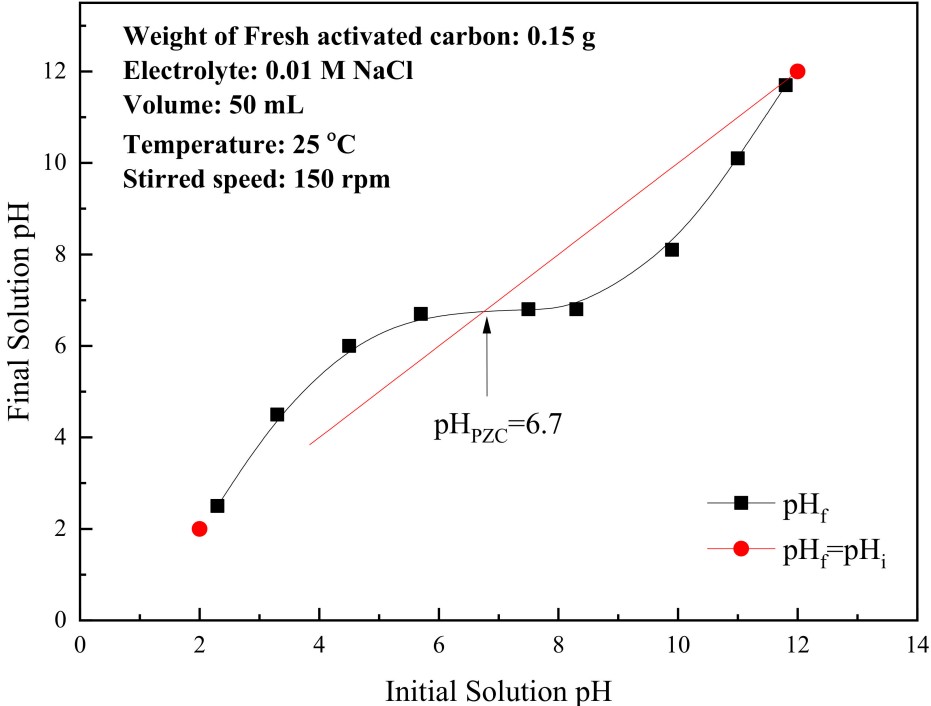

**Figure 3.** Point of zero charge titration for fresh activated carbon.

**Table 1.** Surface elemental characteristics of original and fresh activated carbons.

| Type of Activated Carbon | Element Content (wt. %) | | | | | | |
|---|---|---|---|---|---|---|---|
| | C | O | Mg | P | Si | K | Cl |
| Original Activated Carbon | 89.60 ± 1.66 | 9.26 ± 2.23 | 0.18 ± 0.17 | 0.24 ± 0.25 | 0.24 ± 0.24 | 0.34 ± 0.22 | 0.14 ± 0.12 |
| Fresh Activated Carbon | 92.80 ± 1.15 | 6.40 ± 0.90 | _ a | _ a | 0.08 ± 0.14 | 0.56 ± 0.50 | 0.08 ± 0.14 |

[a]: Not detectable.

**Table 2.** Pore structure characteristics of original, fresh and ultrasound-regenerated activated carbon.

| Activated Carbon | Micropore Volume (cm$^3$/g) | BET Specific Surface Area (m$^2$/g) | Total Pore Volume (cm$^3$/g) | Mean Pore Diameter (nm) |
|---|---|---|---|---|
| Original Activated Carbon | 243.06 ± 0.36 | 1057.9 ± 1.59 | 0.423 ± 0.001 | 1.6 ± 0.002 |
| Fresh Activated Carbon | 250.57 ± 0.38 | 1090.6 ± 1.64 | 0.435 ± 0.001 | 1.6 ± 0.002 |
| Activated Carbon after First Regeneration Cycle | 227.65 ± 0.34 | 990.9 ± 1.49 | 0.395 ± 0.001 | 1.6 ± 0.002 |
| Activated Carbon after Second Regeneration Cycle | 227.88 ± 0.34 | 991.9 ± 1.49 | 0.394 ± 0.001 | 1.6 ± 0.002 |
| Activated Carbon after Third Regeneration Cycle | 223.86 ± 0.34 | 974.4 ± 1.46 | 0.385 ± 0.001 | 1.6 ± 0.002 |
| Activated Carbon after Fourth Regeneration Cycle | 217.93 ± 0.32 | 948.5 ± 1.42 | 0.377 ± 0.001 | 1.6 ± 0.002 |

*3.2. Adsorption of Isopropyl Alcohol*

Solution pH is an important factor as it affects both adsorbent and adsorbate. The effects of initial solution pH on IPA adsorption by activated carbon was investigated in this study, as revealed in Figure 4. The pK$_a$ of IPA is reported to be 16.5 which means that, at the tested range, the IPA is present mainly in the protonated form [26]. The optimal IPA adsorption capacity of 32 mg/g was found for the experiments conducted in an aqueous solution at pH 7, closest to the point of zero charge of fresh activated carbon used in this study. In this range, the hydrogen-bonding between the IPA molecule and the oxygen-containing functional groups on the surface of activated carbon is the greatest [27]. Horng and Tseng tested adsorption of IPA by granulated activated carbon and observed an adsorption capacity of 90 mg/g of activated carbon at an IPA concentration of 1500 mg/L. This higher adsorption capacity, however, was achieved in a continuous system with evaporated acetone in air [28]. One study showed that the IPA concentration and the equilibrium capacity is not necessarily proportional, with their IPA concentrations of 13.4, 22.1 and 31.2 mg/L leading to equilibrium capacities of 0.1737, 0.1581 and 0.1469 mg/g, respectively [29].

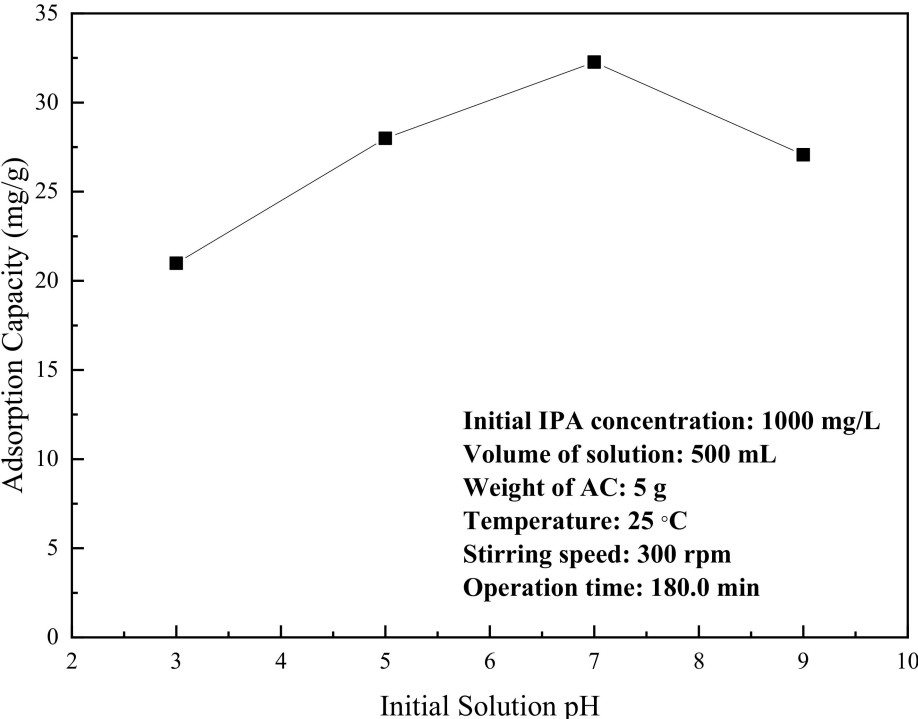

**Figure 4.** The effect of initial solution pH on the adsorption capacity by activated carbon.

### 3.3. Ultrasonic Regeneration

The effect of ultrasonic intensity on IPA desorption from activated carbon is illustrated in Figure 5. An ultrasonic intensity of 32.4 W/cm$^2$ was found to be optimal, followed closely by an intensity of 24.5 W/cm$^2$. Experiments conducted with higher intensities should have more cavitation events occurring, leading to more IPA being desorbed. The intensity of cavitation is primarily dependent on the ultrasonic intensity, which leads to the enhanced breaking of more bonds between IPA and the activated carbon surface. However, a higher ultrasonic intensity can reduce regeneration efficiency as it can damage the activated carbon structure by inducing high-pressure shock waves and high-speed microjets. This effect was measured by comparing the average particle size of activated carbon before and after ultrasonic regeneration. Figure 6 demonstrates this for sonication at 24.5 and 56.4 W/cm$^2$, which showed that the reduction in average particle size of activated carbon increased for desorption conducted with increased ultrasonic intensity. This would explain the decreased regeneration efficiency for desorption experiments conduction with ultrasonic intensities higher than 40 W/cm$^2$. Considering that the desorption with ultrasonic intensities of 24.5 and 32.4 W/cm$^2$ had very similar regeneration efficiencies, it was assumed that an ultrasonic intensity of 24.5 W/cm$^2$ would cause less damage to activated carbon, leading to a higher regeneration efficiency over multiple regeneration cycles. Therefore, an ultrasonic intensity of 24.5 W/cm$^2$ was applied for the majority of ultrasonic regeneration experiments in this study.

The regeneration of IPA-saturated activated carbon by ultrasound was examined at solution temperatures of 25, 35, and 45 °C with temperatures of over 50 °C being avoided to protect the ultrasonic probe. The regeneration efficiency was increased from 70% to 75% in experiments conducted at solution temperatures of 25 and 45 °C. This is likely due to an increase in the breaking of bonds between IPA molecules and the activated carbon surface at higher water temperature, as the desorption process is endothermic in nature [20,30–33]. Moreover, cavitation bubbles are more easily generated as the solution temperature is increased due to a decrease in surface tension and viscosity of the liquid medium.

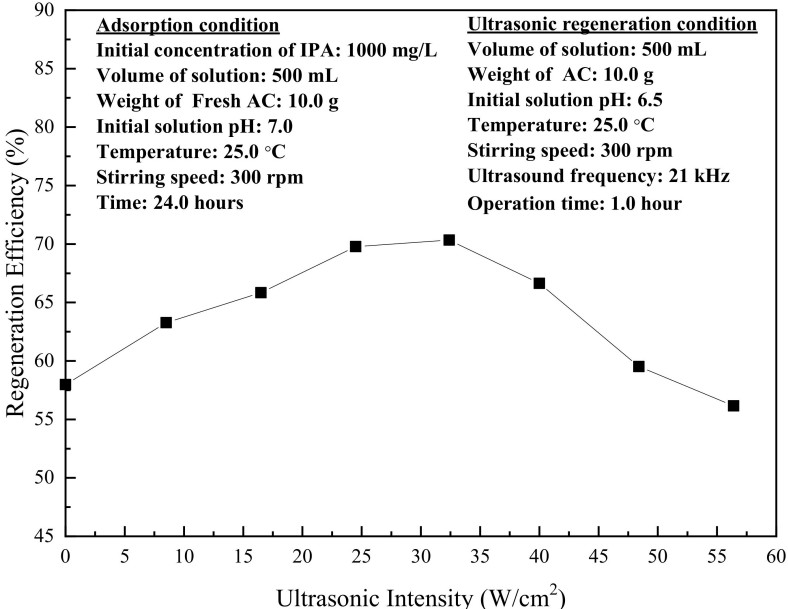

**Figure 5.** Effect of ultrasonic intensity on the regeneration efficiency of exhausted activated carbon.

The effect of ethanol addition on the regeneration efficiency from IPA-saturated activated carbon is shown in Figure 7. The regeneration efficiency of activated carbon is higher for experiments conducted with the addition of 5 and 10% (*v/v*) ethanol. This can be explained due to the decreased tensile stress of aqueous solution by the presence of ethanol. In this study, the optimum ethanol concentration for ultrasonic regeneration of saturated activated carbon was 10% (*v/v*), at which the regeneration efficiency was 83%. However, the regeneration efficiency was decreased for ultrasonic desorption with ethanol concentrations of 15% and 20% (*v/v*), which may be due to a lowering of the cavitation threshold. This causes cavitation bubbles to coalesce into larger and more stable bubbles, thus decreasing the regeneration efficiency [21].

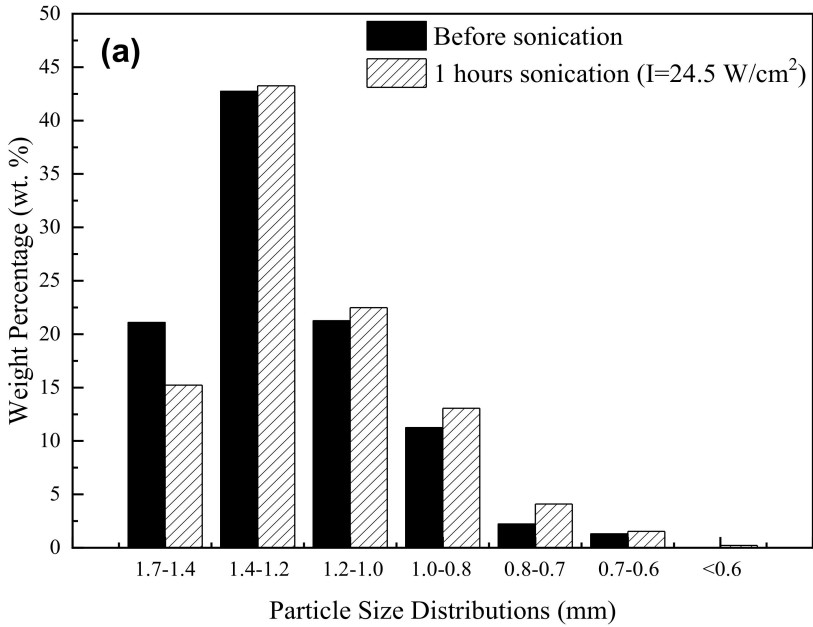

**Figure 6.** *Cont.*

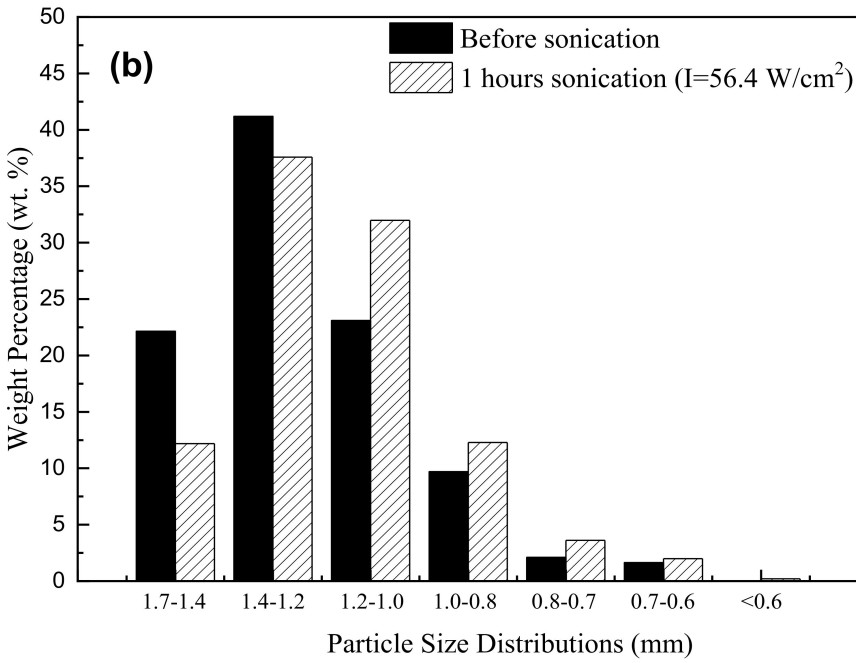

**Figure 6.** Particle size distributions of activated carbon before and after sonication at 24.5 W/cm² (**a**) and 56.4 W/cm² (**b**) for 1.0 h.

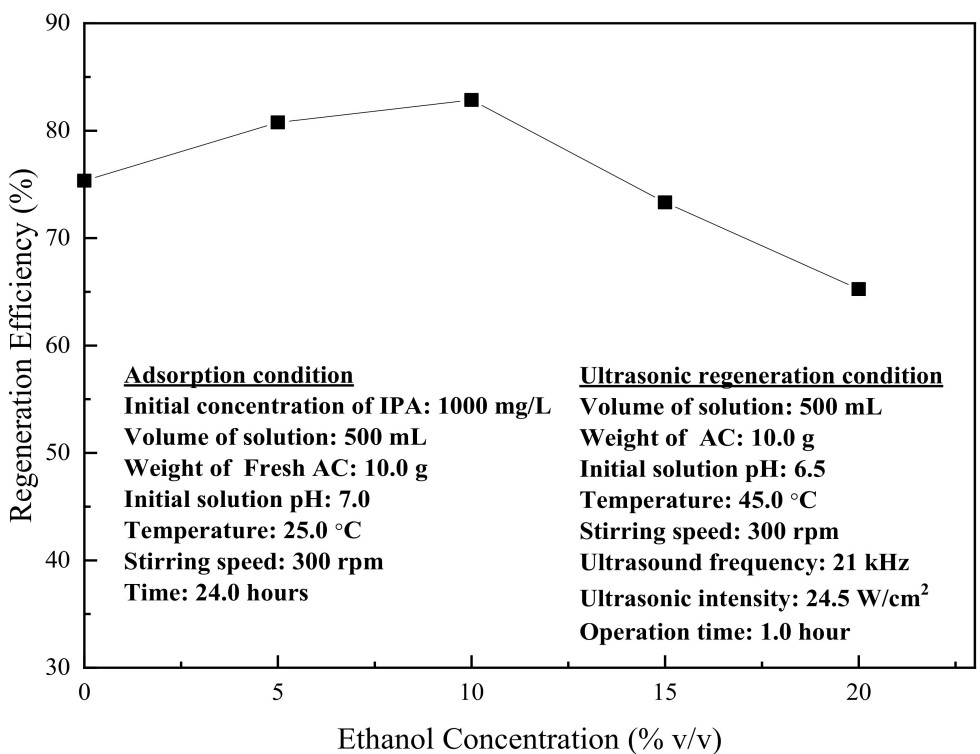

**Figure 7.** Effect of ethanol addition on the regeneration efficiency of exhausted activated carbon.

In this study, it was determined that the optimal operating conditions were an ultrasonic intensity of 24.5 W/cm², a solution temperature of 45 °C and an ethanol concentration of 10% (*v/v*), with a regeneration efficiency of 82.9%. In comparison, Horng and Tseng employed photocatalytic degradation to regenerate IPA-loaded activated carbon and found that, by measuring IPA residue,

>99% of adsorbed IPA was destructed. While initial adsorption capacity was at 90 mg/g, this had dropped to 87 mg/g after one regeneration, a regeneration of over 96% [28]. In tests with other organic solvents, ultrasonic regeneration was found to regenerate 70% of phenol [12] and 64% of trichloroethylene [15] which is lower than the 82% found in this study. The optimal operating conditions were tested in multiple consecutive cycles. Figure 8 shows the reuse potential of regenerated activated carbon over four cycles of adsorption and desorption. The regeneration efficiency gradually decreased from 83% after the first cycle to 81%, 71% and 64% after the second, third and fourth cycles, respectively. In comparison, Horn and Tseng demonstrated that, with their decomposition technique, higher regenerations were achieved of 96% after one regeneration to about 94% after 3 regeneration cycles. While this is a high regeneration of activated carbon, the IPA is destroyed, preventing reuse [28]. This decrease can be attributed to different causes, such as gradual blocking of the porous structure as adsorbate is not completely removed during regeneration. Reduced particle size and increased weight loss of regenerated activated carbon were observed after repeated applications in this study, which is also likely to cause a decrease in adsorption efficiency. A scanning electron microscope (SEM) was used to identify the effect of ultrasound on the morphology of the activated carbon surface. As revealed in Figure 2, SEM images of the original and regenerated activated carbon demonstrated similar surface morphologies before and after ultrasonic regeneration, showing no significant deviation after four operation cycles. The pore structure characteristics of the fresh and regenerated activated carbon over these cycles are presented in Table 2. The results showed that the micropore volume, surface area and total pore volume of regenerated carbon were slightly decreased after four adsorption-desorption cycles. This change in surface morphology is likely another reason for the decreased regeneration efficiency.

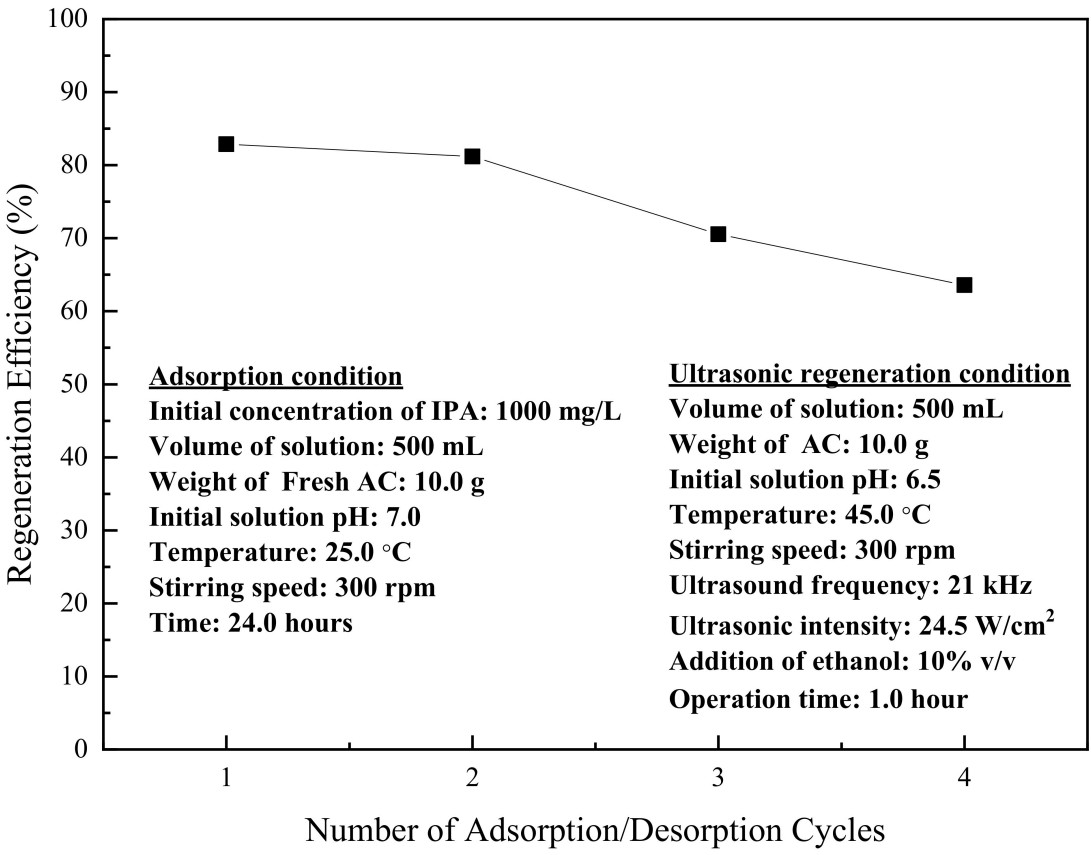

**Figure 8.** Regeneration of exhausted activated carbon over multiple adsorption–desorption cycles.

## 4. Conclusions

The optimal pH tested for IPA adsorption was 7.0. This was very close to the point of zero charge of the activated carbon (6.7), which suggests that hydrogen-bonding between the hydroxyl group on the IPA molecule and the oxygen groups on the surface of activated carbon was a key part in the adsorption mechanism of IPA on activated carbon. A higher ultrasonic intensity led to more IPA being desorbed but also led to greater attrition of the activated carbon, with the optimum intensity at 24.5 W/cm$^2$. The regeneration efficiency increased with temperature up to 45 °C, which was not exceeded in order to protect the ultrasonic probe. Addition of 10% wt. ethanol improved the regeneration efficiency of IPA-saturated activated carbon, but a higher ethanol concentration decreased IPA desorption. Surface morphology of regenerated activated carbon was observed to be similar between these cycles, but surface area and total pore volume decreased. The optimum operating parameters in this study were an ultrasonic intensity of 24.5 W/cm$^2$, a solution temperature of 45 °C and an ethanol concentration of 10% (*v/v*), which led to a regeneration efficiency of 83%. This regeneration efficiency dropped to 81%, 71% and 64% after two, three, and four regeneration cycles. Conclusively, ultrasonic regeneration of IPA-saturated activated carbon was demonstrated to be feasible with regeneration efficiency comparable to current technology.

**Author Contributions:** Conceptualization, Y.K. and H.-Y.H.; methodology, Y.K.; validation, Y.K. and N.M.M.; investigation, H.-Y.H., N.M.M., Y.K. and H.-Y.L.; resources, Y.K.; data curation, H.-Y.H. and Y.K.; writing—original draft preparation, H.-Y.H. and N.M.M.; writing—review and editing, Y.K., H.-Y.L. and N.M.M.; visualization, H.-Y.H. and N.M.M.; supervision, Y.K.; project administration, Y.K. and H.-Y.L.; funding acquisition, Y.K. All authors have read and agreed to the published version of the manuscript.

**Funding:** This research was supported by Grant MOST 108-2218-E-011-005- from the Taiwan Ministry of Science and Technology.

**Conflicts of Interest:** The authors declare no conflict of interest.

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
