# Peer review of "Ultrasonic Regeneration Studies on Activated Carbon Loaded with Isopropyl Alcohol"

_applsci, doi:10.3390/app10217596_

Round 1
Reviewer 1 Report
Page 2, Lines: 56-78. In this part should be presented some examples of ultrasonic regeneration of activated carbon. Also, the new aspect of the work should be highlighted, and the aim of the work presented. Please improve this fragment to show your contribution to the filed.
Page 3, Line: 97. Please indicate which solutions were used for pH adjustment? NaOH? HCl? or others?
Page 4, Line: 127-128. In the paper there is no SEM image of the activated carbon. It should be inserted. Please show the SEM images of the AC original, fresh and after regeneration.
Page 4, Lines: 133-134. The titration data should be presented in a form of graph.
Pages 4-5, Table 2. The standard deviation should be presented.
Page 5, Lines: 151-164. There is no data presentation (kinetics, adsorption isotherm). Is this an earlier study? This should be clearly written and the appropriate citation of the reference should be given. Please rewrite this fragment. The present form is confusing.
Pages 6-7, Figures 3. How was the particles diameter measured?
Page 8, Lines: 221-224. There is no SEM images in the paper!!!!
The results should be compared with other regeneration methods described in the literature.
Author Response
Please see the attachment, thank you for your review
Regards,
N.M. Moed

Reviewer 2 Report
Dear Authors,
Please find attached my comments on your article. It may be interesting but major revision should be made and some parts of manuscript have to be explained or improved.

Author Response

(The authors gave the same response as above.)

Reviewer 3 Report
In Materials and Methods section, authors say: The activated carbon used in this study was made from coconut shells in Vietnam and supplied by Mega Union Technology, Inc.". The authors should indicate the obtention method or a new reference with this respect should be added.
Table 1 shows compositional analyses of original and fresh activated carbons. Three different measurements are shown in it. Do the corresponding measurements correspond to different activated carbons, or different areas of the same sample? should be clarified in the text.
In the text, the authors refer to the SEM technique. The authors say "SEM imaging also revealed that acid washing removed impurities, giving rise to a pore structure with a seemingly smoother surface." Authors should include the corresponding micrographs.
Why the authors say that "Fresh activated carbon exhibited a large specific surface area and total pore volume, likely due to an effect of impurities being removed during the 130 acid washing process"?. This statement needs to be revised.
The same decimals should be shown in Table 2.
The Adsorption of Isopropyl Alcohol section must complete. Different experimental data, as well as some figures in order to develop the study of the adsorption of IPA must be included. This section has scarcely been explained.
For example, authors should explain the next sentence: the optimal IPA adsorption capacity of 32 mg/g was found for the experiments conducted in an aqueous solution at pH 7.
Characterization of activated carbons should be included. For example, some SEM images as well as the BET surface of the activated carbons after the Ultrasonic Regeneration could be included.
How can the authors be sure that the IPA has been desorbed, and the activated carbon has been regenerated?
Author Response

(The authors gave the same response as above.)

Round 2
Reviewer 1 Report
Many thanks to the authors for their comprehensive responses to my comments. The paper is ready to be published in the journal.
Author Response
Thank you again for your comments!
Regards,
N.M. Moed
Reviewer 2 Report
Dear Authors,
Please find below my comments on your revised article. It may be interesting but some experiments have to be performed or repeated.

Author Response
Thank you for your comments.
Response 1 and 2:
I understand both your points and I am already doing this in my personal experiments. Sadly, at this point we are unable to change the experimental setup or do additional tests. I will advise current and future students to keep statistical certainty in mind
Response to point 3:
I had previously rounded the error down but have now increased it to include more decimals.
Kind regards,
N.M. Moed
Reviewer 3 Report
Although the questions suggested by the referee have scarcely explained by the authors, the present manuscript could be consider for publication.
Author Response
Thank you again for your comments. I'm sorry you feel the responses were inadequate, I did try to apply as many changes as possible
Kind regards,
N.M. Moed
Round 3
Reviewer 2 Report
Dear Authors,
I cannot accept the manuscript in this form. Please do at least a few experiments applying real conditions of desorption to check the real advantages of ultrasonic regeneration. It should take not more than 3-4 days and results may save this manuscript.
Kind regards
Author Response
Dear reviewer,
We appreciate the comment but we are not going to be able to make the suggested adjustments. While it would take a few days, we no longer have access to the GC used in this study, the student in charge of this research has graduated and the company that supported this research is already considering a pilot-scale facility based on this. Again, for future research we will be sure to use duplication.
Kind regards and on behalf of the other authors,
N.M. Moed